# NLRP3 Ubiquitination—A New Approach to Target NLRP3 Inflammasome Activation

**DOI:** 10.3390/ijms22168780

**Published:** 2021-08-16

**Authors:** Mahbuba Akther, Md Ezazul Haque, Jooho Park, Tae-Bong Kang, Kwang-Ho Lee

**Affiliations:** 1Department of Applied Life Science and Integrated Bioscience, Graduate School, BK21 Program, Konkuk University, Chungju 27478, Korea; smritymahbuba@gmail.com (M.A.); mdezazulhaque@yahoo.com (M.E.H.); pkjhdn@kku.ac.kr (J.P.); kwangho@kku.ac.kr (K.-H.L.); 2Department of Biomedical Chemistry, College of Biomedical & Health Science, Konkuk University, Chungju 27487, Korea; 3Department of Integrated Bioscience & Biotechnology, College of Biomedical and Health Science, Research Institute of Inflammatory Disease (RID), Konkuk University, Chungju 27478, Korea

**Keywords:** inflammasome, NLRP3, ubiquitination, therapeutic target, inhibitors, post-translational modification

## Abstract

In response to diverse pathogenic and danger signals, the cytosolic activation of the NLRP3 (NOD-, LRR-, and pyrin domain-containing (3)) inflammasome complex is a critical event in the maturation and release of some inflammatory cytokines in the state of an inflammatory response. After activation of the NLRP3 inflammasome, a series of cellular events occurs, including caspase 1-mediated proteolytic cleavage and maturation of the IL-1β and IL-18, followed by pyroptotic cell death. Therefore, the NLRP3 inflammasome has become a prime target for the resolution of many inflammatory disorders. Since NLRP3 inflammasome activation can be triggered by a wide range of stimuli and the activation process occurs in a complex, it is difficult to target the NLRP3 inflammasome. During the activation process, various post-translational modifications (PTM) of the NLRP3 protein are required to form a complex with other components. The regulation of ubiquitination and deubiquitination of NLRP3 has emerged as a potential therapeutic target for NLRP3 inflammasome-associated inflammatory disorders. In this review, we discuss the ubiquitination and deubiquitination system for NLRP3 inflammasome activation and the inhibitors that can be used as potential therapeutic agents to modulate the activation of the NLRP3 inflammasome.

## 1. Introduction

Innate immune signaling plays a protective role against pathogens and mediates an inflammatory state that has been linked with many inflammatory diseases [1]. The effector responses caused by the innate immune system are induced by various pattern recognition receptors (PRRs), which can detect endogenous danger signals (danger-associated molecular patterns—DAMPs) or exogenous signals (pathogen-associated molecular patterns—PAMPs) [1]. In 2002, a novel PRR called inflammasome was identified by Martinon and shown to take part in the effector response against PAMPs or DAMPs [2]. Inflammasomes are multimeric protein complexes that regulate the activation and cleavage of caspase-1, and the processing of pro-IL-1β and pro-IL-18 into their mature forms [2]. Among inflammasomes, nucleotide-binding domain (NLR) and pyrin domain-containing receptor 3 (NLRP3) has been linked with many inflammatory and autoimmune diseases [3,4]. NLRP3 regulates the immune response against a wide range of stimuli that can be found endogenously due to aging [5]. The NLRP3 inflammasome contributes to the pathogenesis of age-associated diseases such as atherosclerosis, type 2 diabetes, and gouty arthritis [6]. Many studies suggest that neuroinflammatory and neurodegenerative diseases such as Alzheimer’s disease [7] and Parkinson’s disease [8] are linked with the NLRP3 inflammasome. The dysregulation of NLRP3 inflammasome activation leads to several autoimmune diseases such as multiple sclerosis (MS) [9] and experimental autoimmune encephalomyelitis (EAE) [10]. The NLRP3 inflammasome is also linked with various cancers such as breast cancer, colon cancer, gastrointestinal cancer, and melanoma [11,12]. The molecular mechanisms of NLRP3 inflammasome activation and how they are linked with disease pathogenesis are emerging topics of research interest. The post-translational modifications (PTMs) of the NLRP3 inflammasome, such as ubiquitination and phosphorylation, are critical for NLRP3 inflammasome activation [13,14,15,16].

A plethora of studies has established that ubiquitination is indispensable for controlling inflammasome activation [13,17,18]. The ubiquitin system fundamentally controls cellular protein homeostasis by lysosomal or proteasomal degradation [19,20]. However, the ubiquitin system also maintains other cellular outcomes such as signal transduction, protein–protein interaction, and alteration of subcellular localization [21,22]. Additionally, components of ubiquitination such as E3 ligases and deubiquitinases (DUBs) are successful therapeutic targets in cancer [23].

A comprehensive understanding of the ubiquitin system’s regulation of the NLRP3 inflammasome cascade may contribute to targeted therapeutic interventions for NLRP3 inflammasome-mediated diseases. In this review, we focus on NLRP3 inflammasome activation regulation by the ubiquitination system and the potential pharmacological inhibitor of NLRP3 regulation through the ubiquitin system, which could have therapeutic implications for NLRP3 inflammasome-associated diseases.

## 2. NLRP3 Inflammasome Activation

Inflammasomes are multiprotein signaling complexes composed of sensor proteins, adaptor proteins, and caspases [2,24,25,26]. Inflammasomes control the inflammatory response and coordinate host defenses against infection. In response to pathogenic microorganisms and danger signals, inflammasome complexes are assembled by pattern-recognition receptors (PRRs) followed by an adaptor molecule, ASC, and an inflammatory caspase, caspase-1 [27]. The activation of caspase-1 produces mature forms of pro-inflammatory cytokines, such as IL-1β and IL-18, and cleaves gasdermin D (GSDMD), which leads to pyroptotic cell death [26]. Inflammasome formation requires PRRs to sense the pathogenic or danger signals. To date, five PRRs have been reported to form inflammasome complexes: three from the NOD-like receptor (NLR) family (NLRP1, NLRP3, and NLRC4), Pyrin, and AIM2. In addition, it has been reported that other inflammasomes of the NLR family (NLRP6, NLRP7, NLRP12) and the PYHIN (IFI16) family form inflammasome complexes without any defined components [28,29].

The NLRP3 receptor, a member of the NLR family, is a tripartite protein that contains three domains: an amino-terminal pyrin (PYD) domain; a NACHT domain, which is vital for NLRP3 self-association and function due to its ATPase activity; and a carboxy-terminal leucine-rich repeat (LRR) domain, which is capable of auto-inhibition by folding back onto the NACHT domain [30,31,32]. To form an inflammasome complex, the NLRP3 inflammasome needs a sensor NLRP3, an adaptor apoptosis-associated speck-like protein containing a CARD (ASC or also known as PYCARD), and an effector caspase-1 [33]. ASC consists of two domains: an amino terminal PYD domain and a carboxy-terminal caspase recruitment domain (CARD) [34]. A full-length caspase-1 is composed of one amino-terminal CARD domain and two catalytic domains, the latter composed of a central large catalytic (p20) domain and a carboxy-terminal catalytic small subunit (p10) domain [31]. There are two proposed signals for the activation of the NLRP3 inflammasome—the priming or first signal, and the activation or second signal [31,35].

### 2.1. Priming Signal (Signal 1) of NLRP3

Diverse exogenous and endogenous signals, such as damaged-associated molecular patterns (DAMPs) released from damaged or dying cells; pathogen-associated molecular patterns (PAMPs) released from microbial infections; PRRs, such as Toll-like receptors (TLR) and NOD-like receptor (NLR) ligands [36]; TNF-α [35]; IL-1 [37]; and toxins (MPTP) [38] all act as priming or first signals [39]. The basal level of NLRP3 in macrophages is not enough in quantity for NLRP3 inflammasome activation. Therefore, priming signals are essential for the increase in NLRP3 transcription, which licenses NLRP3 for the activation step [35]. Pro ILIβ also does not express steadily in resting macrophages that are rather inducible by PAMPs [35,40]. After sensing a priming signal, a sequence of events is induced, including translocation of transcription factor NF-κB to the nucleus and its activation, which results in transcription of NLRP3 and pro-IL-1β [35].

### 2.2. Activation Signal (Signal 2) of NLRP3

Mitochondrial reactive oxygen species (mtROS), K+ efflux, Ca+ flux, different misfolded proteins, lysosomal disruption, adenine triphosphate (ATP), pore-forming toxins, crystalline substances, particulate matter, and viral RNA [41] are the most common activators or second signals of NLRP3 inflammasome activation [6,42,43]. These signals initiate the activation, assembly, and inflammasome complex formation [44,45].

Following an activation signal, NLRP3 molecules are oligomerized through the NACHT domain homotypic interactions [39]. The NLRP3 oligomer recruits ASC through a PYD–PYD interaction and promotes ASC speck formation through the assembly of multiple ASC filaments. ASC then recruits caspase-1 through a CARD–CARD interaction and activates caspase-1 through proximity-induced self-cleavage between p20 and p10 of caspase-1 [46]. Studies suggest that a serine-threonine kinase, NIMA-related kinase 7 (NEK7), interacts with NLRP3 and form oligomers that are essential for ASC speck formation and caspase-1 activation [47,48]. A recent study showed that this NEK7−NLRP3 interaction can be regulated by the phosphorylation of NEK7. The phosphorylation of NEK7 occurs at Ser204 by polo-like kinase 4 (PLK4) upon LPS stimulation [49]. Since the phosphorylation attenuates the association of NEK7 with NLRP3, it might serve to restrain inflammasome activation [49]. Following cleaved mediated activation of caspase-1, heterotetrameric caspase-1 proteolytically cleaves the pro-form of IL-1β and IL-18 to yield mature IL-1β and IL-18. Mature IL-1β and IL-18 are then released through the non-classical secretion pathway [50,51]. In addition, activated caspase-1 cleaves and activates another protein—gasdermin D (GSDMD). After activation, GSDMD translocates to the plasma membrane and forms pores that contribute to the release of IL-1β and IL-18 into the extracellular space. Subsequently, pore formation of GSDMD induces a pro-inflammatory form of cell death known as pyroptosis [52].

## 3. Post-Translational Modification (PTM) of NLRP3

Post-translational modification (PTM) plays a crucial role in the activation of the NLRP3 inflammasome upon priming stimuli. Before priming, NLRP3 remains in an inactive conformation [31]. NLRP3 activation may be controlled through combinations of different post-translational modifications. The phosphorylation of NLRP3 has been found to both induce and inhibit NLRP3 activation depending on the site of phosphorylation and stage of NLRP3 activation [15,53,54,55]. TLR-dependent phosphorylation of NLRP3 prepares it for the following stimulation [54]. Ubiquitination and deubiquitination also play crucial roles in the degradation or activation of the NLRP3 inflammasome [17,56]. TLR or IL-1R triggers TRIM31 and MARCH 7 ubiquitinase, which targets the degradation of NLRP3 through ubiquitination [57], whereas priming activates BRCC3 deubiquitylates NLRP3, which promotes its homo-oligomerization [17]. Phosphorylation can alter the ubiquitination of NLRP3; for example, the phosphorylation of NLRP3 at S194 is critical for NLRP3 deubiquitination and self-association [54], and the phosphorylation of NLRP3 at S291 facilitates K48- and K63-linked polyubiquitination and inhibits NLRP3 activation by promoting its degradation [54]. Notably, PTM plays an important role in NLRP3 inflammasome activation; thus, targeting post-translational modifications of the NLRP3 inflammasome is a potential therapeutic avenue by which to treat NLRP3-associated inflammatory disorders.

## 4. Mechanism of Ubiquitination

Ubiquitination is a form of post-translational modification of proteins by the conjugation of a ubiquitin protein to the substrate proteins. Ubiquitin (Ub) is a 76 amino acid protein and is usually conjugated to the lysine residue of substrate proteins through an isopeptide bond [58]. Ubiquitination occurs through a sequential activity of three enzymes, namely, ubiquitin-activating enzymes (E1), ubiquitin-conjugating enzymes (E2), and ubiquitin-ligating (E3) enzymes [59]. The cascade of reaction starts with the activation of the C-terminus of Ub by the E1 enzyme to form a thioester bond. The Ub is then transferred to the active cysteine site of the E2 enzyme by trans-thioesterification. Finally, Ub is transferred to the substrate protein by E3 ligase and facilitates isopeptide bond formation among the C-terminal glycine of the Ub and the lysine residue of the substrate protein [58]. E3 ligases can be classified into three major categories: really interesting new gene (RING), homology to E6AP C terminus (HECT), and RING between RING (RBR). Among them, the RING family can bind to both E2~Ub thioester and the substrate concurrently and can directly transfer Ub to the substrate. HECT and RBR functions through a two-step reaction: Ub is first transferred from E2 to an active-site cysteine site in the E3, and from there is transferred to the substrate [60].

Only two families of E1 have been discovered, whereas there are around 40 and 600 known families for E2 and E3 enzymes, respectively [61,62,63]. Both E2 and E3 enzymes are responsible for the specificity of the ubiquitination to its substrate [64]. Deubiquitinating enzymes (DUB) are responsible for the removal of ubiquitin chains from proteins by the cleavage of isopeptide bonds among lysine residue on the protein and the Ub C-terminus glycine [65]. Around 100 deubiquitinating enzymes (DUB) have been discovered that can potentially remove Ub from substrate proteins [66]. Broadly, these DUBs have been classified into seven major classes: (1) ubiquitin-specific proteases (USPs), (2) ubiquitin C-terminal hydrolases (UCHs), (3) ovarian-tumor proteases (OTUs), (4) Machado–Joseph disease protein domain proteases (MJD), (5) monocyte chemotactic protein-induced protein (MCPIP), (6) MIU-containing novel DUB family (MINDY), and (7) JAMM/MPN domain-associated metallopeptidases (JAMMS). Among them, JAMMS is the only zinc metalloprotease group and the others are in the thiol protease group [67,68].

During ubiquitination, the C-terminus of ubiquitin is conjugated to the lysine (K) residue of its substrate. Based on the number of ubiquitins and the attachment type, different types of ubiquitination can occur. If ubiquitination occurs in a single residue, it is called monoubiquitination; if multiple ubiquitination occurs in the same protein but at different sites, it is known as multimonomeric ubiquitination [58]; and when multiple ubiquitination occurs at a single site, it is referred to as poliubiquitination. Ubiquitin itself contains seven lysine residues (K6, K11, K27, K29, K33, K48, K63) and one methionine [58]. During the formation of a polymeric chain, the ubiquitin lysine residue attached to the substrate serves as a platform to conjugate additional ubiquitin residues. If the same residue is modified, it causes homogenous polymeric chain formation. However, additional ubiquitins can conjugate to any of these eight ubiquitin residues to form mixed polyubiquitin chains [58,69].

## 5. Regulation of NLRP3 Inflammasome Activation by Ubiquitination

Ubiquitination is crucial to control NLRP3 inflammasome activation [17,56]. Since the discovery of the potential role of ubiquitination in NLRP3 activation, NLRP3 ubiquitination has been studied widely and the involvement of several E3 ligases and DUBs has been confirmed.

### 5.1. Molecules Involved in the Ubiquitination of NLRP3

#### 5.1.1. E3 Ligases Regulating NLRP3

Ubiquitination of a protein requires E1, E2, and E3 ligases, and so far, E3 ligases have mainly been reported to be linked with NLRP3 inflammasome activation [70]. However, some E3 ligases prevent NLRP3 inflammasome activation and work as a negative regulator of NLRP3 inflammasome activation [71,72]. NLRP3 is maintained at low levels within inactivated cells, which may be a result of its degradation by K48-linked Ub chain-dependent proteasomal degradation and K63-linked Ub chain-dependent autophagic degradation [56,73]. This is thought to be the case because in inactivated cells, NLRP3 is highly ubiquitinated by both K63- and K48-linked polyubiquitin chains [56]. To date, several E3 ligases have been identified to regulate NLRP3 inflammasome activation (Table 1, Figure 1).

MARCH7

A study showed the inhibiting effect of dopamine, a neurotransmitter, on NLRP3 inflammasome activation by NLRP3 protein degradation [57]. In the study, they identified an E3 ligase—MARCH7—by mass spectrometry that ubiquitinates NLRP3 and promotes its proteasomal degradation. Dopamine initiates dopamine D1 receptor (DRD1) signaling, which in turn facilitates cyclic adenosine monophosphate (cAMP) production. Then, cAMP binds to NLRP3, leading to K48-linked ubiquitination of NLRP3 initiated by MARCH 7. The ubiquitinated NLRP3 becomes self-associated to form aggregates and is finally targeted for autophagy-mediated degradation. Therefore, the E3 ligase MACH7 inhibits NLRP3 inflammasome activation through the induction of ubiquitination and degradation of NLRP3 [57].

SCF-FBXL2

Another E3 ligase, SCF-FBXL2, has been reported to negatively regulate NLRP3 inflammasome activation [74]. It is well acknowledged that LPS priming induces an increase in NLRP3 levels in cells [35]. A study showed that SCF-FBXL2 targets K689 and initiates NLRP3 protein degradation, which keeps the NLRP3 protein at low levels in inactivated cells. However, LPS priming suppresses the ubiquitin-mediated NLRP3 degradation [74]. LPS priming induces FBXO3-mediated ubiquitination and degradation of FBXL2, which increases the NLRP3 levels and enhances the NLRP3 inflammasome activation [74]. 

TRIM31

TRIM31, an E3 ligase, has been demonstrated to regulate the NLRP3 inflammasome activation as a feedback suppressor [71]. TRIM31 directly binds to the PYD domain of NLRP3 via the N-terminal RING domain and induces K48-linked polyubiquitination. This causes NLRP3 proteasomal degradation in both activated and inactivated cells. In addition, a deficiency of TRIM31 attenuates dextran sodium sulfate (DSS)-induced colitis through NLRP3 inflammasome activation [70]. This demonstrates the inhibitory role of TRIM31 on the NLRP3 inflammasome activation in pathological conditions. 

ARIH2 (Ariadne Homolog 2)

A previous study demonstrated that an E3 ligase, ARIH2, which is derived from the really interesting new gene (RING) in between RING (RBR) E3 ligase family, suppresses NLRP3 inflammasome activation [77]. ARIH2 can interact with the NACHT domain (aa220-575) of NLRP3 through its RING 2 domain to induce K48- and K63-linked polyubiquitination. Although ARIH2-mediated K48-linked ubiquitination was not involved with any proteasomal degradation of the NLRP3, the deletion of endogenous ARIH2 suppresses NLRP3 ubiquitination and enhances NLRP3 activation [72]. Moreover, the depletion of ARIH2 in cells completely halts NLRP3 ubiquitination, which suggests a predominant role of ARIH2 in NLRP3 ubiquitination [72]. ARIH2 acts as a fine-tuning regulator of aggregate-prone proteins [78]; therefore, it is possible that ARIH2 regulates NLRP3 inflammasome aggregation and negatively regulates NLRP3 inflammasome assembly and activation [72]. 

Cullin 1

Cullin 1 is an E3 ubiquitin ligase and part of the Skp1-Cullin-1-F-boxE3 ligase complex [79,80]. The C-terminal of Cullin 1 interacts directly with the PYD domain of NLRP3 and promotes its K63-linked ubiquitination [75]. The elevation of K63-linked ubiquitination by Cullin 1 prevents the formation of NLRP3 inflammasome assembly. However, in the presence of an NLRP3 inflammasome activator such as ATP, Cullin 1 dissociates from the NLRP3, which promotes NLRP3 inflammasome activation [75].

Cbl-b and RNF-125

Casitas-B-lineage lymphoma protein-b (Cbl-b), a RING-finger E3 ubiquitin ligase, negatively regulates NLRP3 inflammasome activation [81,82]. Upon NLRP3 inflammasome stimulation, Cbl-b binds to the K63-linked ubiquitin chains in the LRR domain of NLRP3 by its ubiquitin-associated domains (UBA). This binding leads to K48-linked ubiquitination at the K496 site in the NBD domain of NLRP3 and induces its proteasomal degradation [18]. K63-linked polyubiquitination in the LRR domain of NLRP3, which is indispensable for Cbl-b recruitment, is initiated by RNF125, another RING-finger E3 ubiquitin ligase [18,83]. Thus, RNF125 and Cbl-b ubiquitinate NLRP3 for K63- and K48-linked polyubiquitination, respectively, preventing NLRP3 inflammasome activation [18].

Pellino-2

Pellino-2, an E3 ubiquitin ligase [84], regulates NLRP3 inflammasome activation both positively and negatively [84]. LPS priming leads to an association of pellino-2 with NLRP3 and promotes K63-linked polyubiquitination of NLRP3. This ubiquitination may facilitate NLRP3 inflammasome activation [76]. However, Pellino-2 can also inhibit NLRP3 inflammasome activation. In wild-type bone marrow-derived macrophages (BMDM), Pellino-2 ubiquitinates IL-1R associated kinase 1 (IRAK1), which is required for rapid activation of NLRP3 [85]. The ubiquitination of IRAK1 can inhibit NLRP3 inflammasome activation. Since ubiquitinating IRAK1 limits its ability to associate with NLRP3, this inhibits NLRP3 activation [76].

TRAF6

TRAF6 is associated with the non-transcriptional priming of the NLRP3 inflammasome via its ubiquitin E3 ligase activity [86]. TRAF6 is known for its role in NF-kB-induced transcription of NLRP3. However, it has been shown that TRAF6 facilitates NLRP3 oligomerization and an ASC–NLRP3 interaction [86]. This indicates that TRAF6 is a positive regulator for TLR-mediated NLRP3 inflammasome activation.

HUWE1

HUWE1 is an E3 ubiquitin ligase [87] that interacts with the NACHT domain of NLRP3 via its BH3 domain [88]. The interaction of HUWE1 and NLRP3 causes K27-linked polyubiquitination of NLRP3, which favors NLRP3 assembly, ASC speck formation, and caspase-1 activation. Therefore, HUWE1 works as a positive regulator of the NLRP3 inflammasome. In addition, HUWE1 regulates AIM2 and NLRC4 inflammasome activation by the same mechanism [88].

TRIM33

Tripartite motif-containing 33 (TRIM33) is another positive regulator of NLRP3 inflammasome activation [89]. The interaction of the cytosolic dsRNA sensor DHX33 with NLRP3 induces inflammasome activation by the help of an E3 ligase, TRIM33. TRIM33 can initiate the K63-linked ubiquitination of DHX33 at K218, which results in a DHX33–NLRP3 inflammasome complex and activation of the NLRP3 inflammasome [90]. Additionally, NLRP3 inflammasome activation was prevented by the depletion of TRIM 33 in macrophage cells, which highlights the essential role of TRIM33 in the initiation of NLRP3 inflammasome activation in response to cytosolic RNA stimulation [89].

β-TrCP1

β–transducin repeat-containing E3 ubiquitin protein ligase 1 (β-TrCP1) is an E3 ligase from the F-box protein family and has two paralogs, β-TRCP1 and β-TRCP2 [91], whose role in ubiquitination has long been established [92]. A recent study revealed that β-TRCP1 increases K27-linked ubiquitination and the overall polyubiquitination of NLRP3 by directly binding to NLRP3 [93]. The study also found that β-TRCP1 promotes K27-linked ubiquitination in K380 of the NACHT domain of NLRP3, which in turn promotes its proteasomal degradation to act as a negative regulator for NLRP3 inflammasome activation [93].

TRIM24

Tripartite motif-containing 24 (TRIM24) belongs to the tripartite motif (TRIM) family of proteins, which contains RING-finger domain-contributing E3 ligases [71]. TRIM24 is a transcriptional factor that negatively regulates inflammatory responses [94]. A recent study showed that TRIM24 interacts with NLRP3 in such a way that it leads to an increase in NLRP3 ubiquitination [95]. Moreover, TRIM24 deficiency promotes NLRP3 inflammasome activation. Therefore, E3 ligase TRIM24 may negatively regulate NLRP3 inflammasome activation via its ubiquitination [95].

Parkin

Parkin, an RBR E3 ligase, has been reported to regulate NLRP3 inflammasome activation [94]. A study found that Parkin regulates NLRP3 inflammasome activation negatively, as Parkin-deficient cells have increased NLRP3 activation [96]. Mechanistically, Parkin induces the ubiquitin-modifying enzyme anti-apoptotic signaling protein 20 (A20), which attenuates the priming effect on the NLRP3 inflammasome by suppressing NF-κB activation [96,97].

Ubc13

Ubc13 is an E2 enzyme that regulates the assembly of K63-linked polyubiquitin chains specifically [98]. Additionally, Ubc13 transduces the NF-κB signal by interacting with TNF receptor-associated factor 6 (TRAF6) [99]. A recent study found that Ubc13 acts as a positive regulator for NLRP3 inflammasome activation, as Ubc13 deficiency significantly reduced NLRP3 inflammasome activation [100]. The study also revealed that Ubc13 interacts with NLRP3 to induce K63-linked polyubiquitination at K565 and K687 sites. Therefore, Ubc13 works as a positive regulator of NLRP3 inflammasome activation by promoting K63-linked polyubiquitination of NLRP3 [100].

#### 5.1.2. Deubiquitinase Regulating NLRP3

Deubiquitinases (DUBs) play a crucial role in controlling the level of NLRP3 inflammasome activation along with E3 ubiquitin ligases. Several DUBs have been shown to regulate NLRP3 inflammasome activation (Figure 1, Table 2) [17,101,102].

BRCC3

BRCA1/BRCA2-containing complex subunit 3 (BRCC3) is a DUB in the JAMM domain-containing Zn2+ metalloprotease that also takes part in the formation of the BRCC36 isopeptidase complex (BRISC) [103,104]. BRCC3 was identified as a DUB through screening the DUB library expression that can regulate NLRP3 inflammasome activation [105]. BRCC3 can directly bind to NLRP3 and favors NLRP3 inflammasome activation through its deubiquitination. The G5 BRCC3 inhibitor blocks NLRP3 inflammasome activation, which verifies the role of BRRC3 as a positive regulator of NLRP3 inflammasome activation [17]. A later study identified an upstream molecular mechanism for the regulation of NLRP3 through BRCC3 [106]. The study identified another component of the BRISC complex, Abraxas brother 1 (ABRO1), which also regulates NLRP3 inflammasome activation [106]. The study revealed that upon LPS priming, ABRO1 binds to NLRP3 and recruits BRISC to promote K63-dependent deubiquitination of NLRP3 [106] and that a deficiency of BRCC3/ABRO1 attenuates NLRP3-associated inflammatory diseases such as peritonitis or sepsis [107].

**Table 2 ijms-22-08780-t002:** Regulation of NLRP3 by DUB.

Name	Regulation of Inflammasome Activation	Type of UB Linkage	Ubiquitination Site	Interaction	Stage of NLRP3 Activation	Reference
BRCC3	Positive	Not identified	LRR	Not identified	Priming	[17]
USP7/USP47	Positive	Not identified	Not identified	Not identified	Priming	[108]
UAF1	Positive	K48	Not identified	LRR, NACHT	Priming	[101]
STAMBP	Negative	K63	Not identified	Not identified	Upon endotoxin stimulation	[109]

USP7/USP47

USP7 and USP47 are members of the ubiquitin-specific protease DUB family [110,111]. USP7 cleaves several ubiquitin-linked chains such as K11, K63, and K48 [112], whereas USP47 is known for its enzymatic activity as a deubiquitinase and DNA repair [113]. USP47 contains the adjacent structure of USP7 [111]. Both chemical inhibition and knockout of USP7 and USP47 in macrophages demonstrated a blockage of transcription-independent NLRP3 inflammasome activation through the prevention of ASC oligomerization and speck formation [108]. Additionally, upon NLRP3 inflammasome stimulation, USP7/USP47 activity was increased, which suggests a concurrent post-translational modification. However, it is not clear whether USP7 and USP47 directly contribute to NLRP3 deubiquitination or regulate the process somewhere upstream of NLRP3 inflammasome action [108].

UAF1

Ubiquitin specific peptidase 1 (USP1)-associated Factor 1 (UAF1) [102] is a component of three DUB enzyme complexes, namely, UAF1/USP1, UAF1/USP12, and USP1/USP46 [102]. The cellular NLRP3 level is kept in check by the K48 ubiquitination-dependent proteasomal degradation [71,72]. A recent study identified the UAF1/USP1 deubiquitinase complex that removes K48-linked ubiquitination of NLRP3 [101], which prevents the proteasomal degradation of NLRP3 and increases NLRP3 mRNA and protein, which facilitates NLRP3 inflammasome activation [101]. Additionally, the study also found out that the UAF1/USP12 and UAF1/USP46 complexes increase p65 expression, which promotes NF-κB activation and increases NLRP3 and IL-1β expression levels. Thus, UAF1 deubiquitinase complexes work as a positive regulator for NLRP3 inflammasome activation. In support of these results, the UAF1 deficiency both in vitro and in vivo further showed a decrease in NLRP3 inflammasome activation and IL-1β secretion [101]. 

STAMBP

Signal transducing adaptor molecule-binding protein (STAMBP) is an endosome-resident deubiquitinase in the Jab/MPN metalloenzyme (JAMM) family [114]. STAMBP regulates the endolysosomal cellular trafficking of ubiquitinated proteins such as NALP7 [115]. A recent study found that STAMBP negatively regulates NLRP3 inflammasome activation [109]. Moreover, STAMBP deficiency increases the IL-1β gene expression and cytokine level in response to LPS, and showed an increase in ASC speck and active caspase-1 levels. In addition, STAMBP knockout cells showed an increase in K63-linked NLRP3 polyubiquitination upon endotoxin exposure. Therefore, the deubiquitination of NLRP3 by STAMBP acts as a negative regulator for NLRP3 inflammasome activation [109]

#### 5.1.3. Ubiquitinase and DUB-Independent Regulation of the NLRP3 Ubiquitination

HDAC6

Histone deacetylase (HDAC6) exhibits ubiquitin-binding activity and can transport ubiquitinated protein via microtubules [116]. A previous study revealed that HDAC6 can associate with ubiquitinated NLRP3 by its ubiquitin-binding domain and negatively regulates NLRP3 inflammasome activation. Moreover, treatment of a deubiquitinase inhibitor PR619 caused an increased association between HDAC6 and NLRP3 [117]. These results indicate that HDAC6 may negatively regulate NLRP3 inflammasome activation through a direct association with ubiquitinated NLRP3 [117].

### 5.2. Regulation of ASC by Ubiquitination

#### 5.2.1. E3 Ligases That Regulate ASC

LUBAC

Linear ubiquitin assembly complex (LUBAC), which consists of HOIP, HOIL-1L, and SHARPIN (SHANK-associated RH domain interactor), forms Methionine 1 (Met1)-linked linear ubiquitin chains to the substrate proteins that regulate the classical activation of NF-κB [118]. A previous study demonstrated that LUBAC is essential for the linear ubiquitination of ASC in Met1 [119]. Moreover, this study showed that a deficiency of HOIL-1, a component of LUBAC, results in the suppression of IL-1β secretion both in vitro and in vivo, independent of transcriptional regulation [119]. Another study demonstrated that a SHARPIN deficiency hinders NLRP3 inflammasome activation in mouse macrophages [120]. A SHARPIN deficiency, unlike a HOIL deficiency, results in an impaired NF-κB pathway, which suggests that it has a role in the transcription of the NLRP3 inflammasome component [119,120]. These studies suggest that linear ubiquitination of ASC by LUBAC plays an important role in the activation of the NLRP3 inflammasome.

TRAF3

Severe acute respiratory syndrome coronavirus (SARS-CoV) open reading frame (ORF3) is known to activate the NLRP3 inflammasome by regulating ASC ubiquitination. ORF3 has been found to interact with TRAF3 and ASC, which leads to K63-linked polyubiquitination of ASC [121]. It was also found that ubiquitination of ASC occurs in a TNF receptor-associated factor (TRAF3)-dependent manner [121]. In another study, it was found that upon vesicular stomatitis virus infection, ASC goes through K63-dependent polyubiquitination at K174 in a mitochondrial antiviral signaling protein (MAVS)-dependent manner [122]. Moreover, TRAF3 acts as an E3 ligase for the K63 ubiquitination of ASC (Table 3). Deficiencies of TRAF3 and MAVS showed an impairment of inflammasome activation caused by ASC speck formation, which indicates that K63-linked ubiquitination is crucial for inflammasome activation [122].

TRAF6

TNF receptor-associated factor 6 (TRAF6)-mediated ASC ubiquitination plays a role in the suppression of inflammasome activation rather than induction. Upon stimulation with far-infrared, macrophages undergo K63-linked polyubiquitination in ASC by TRAF6 (Table 3) [123,125]. The polyubiquitination of ASC suppresses NLRP3 inflammasome activation due to the induction of the autophagic degradation of NLRP3 [123].

#### 5.2.2. DUBs That Regulate ASC

Although the PTM of ASC has not yet been fully clarified, a study identified that ubiquitin-specific peptidase 50 (USP-50), a DUB, may be involved in regulating NLRP3 inflammasome activation through the deubiquitination of ASC (Table 3) [126]. It was confirmed that USP50 deubiquitinates K63-linked polyubiquitination of ASC proteins. In addition, USP-50 knockdown in macrophages significantly impairs NLRP3 inflammasome activation [124].

### 5.3. Caspase-1 Ubiquitination

Whether ubiquitination of caspase-1 is required to mediate the NLRP3 inflammasome complex is not yet clear, but some studies have suggested the involvement of the ubiquitination of caspase-1 [127,128]. ZIKA virus infection induced inflammasome activation by regulating caspase-1 ubiquitination [127]. It was shown that a non-structural protein of ZIKA virus, NS1, recruits USP8, a deubiquitinase (DUB), and targets caspase-1 to remove K11-linked ubiquitin chains at K134 of caspase-1 [129]. Since K11-linked polyubiquitination of caspase-1 is responsible for the proteasomal degradation of caspase-1 and DUB, the recruitment of USP8 can reverse caspase-1 degradation by increasing its stability [127]. 

Cellular inhibitors of apoptosis proteins (cIAPs) can also regulate inflammasome activation through caspase-1 ubiquitination [130]. cIAP1/cIAP2 and the adapter protein TRAF2 potentially interact with caspase-1 and promote its K63-linked polyubiquitination, which positively regulates NLRP3 inflammasome activation. Moreover, a deficiency of cIAP1/cIAP2 leads to impaired activity of caspase-1 and subsequent NLRP3 inflammasome activation in mice [130]. However, a later study obtained a conflicting result that the inhibition of cIAP1, cIAP2, and XIAP activated NLRP3 inflammasome and IL-1β production in a receptor-interacting protein kinase (RIPK3)-dependent manner [131], which indicates that cIAP might have an inhibitory effect on NLRP3 inflammasome activation induced by a non-classical pathway.

### 5.4. IL-1β Ubiquitination

NLRP3 inflammasome activation eventually results in the cleavage of pro-IL-1β and its secretion from the cells [2]. IL-1β has also been linked with polyubiquitination and proteasomal degradation in bone marrow-derived dendritic cells (BMDC) [132]. An E2 ubiquitin ligase, UBEL2L3, promotes K48-linked ubiquitination of pro-IL1β to drive its proteasomal degradation, thus decreasing the amount of cleaved IL-1β [128]. Another study focused on the impact of A20, a deubiquitin-modifying enzyme, on inflammasome activation [133]. This study demonstrated that the depletion of A20 in macrophages results in spontaneous IL-1β processing and release in response to priming activation. Further mechanistic studies showed that ubiquitination at K133 of mouse pro-IL-1β supports the cleavage of pro-IL-1β and A20 limits this proteolytic cleavage through the restriction of pro-IL-1β ubiquitination [133]. 

In line with this notion, another deubiquitinase, POH1, was also reported to restrict NLRP3 inflammasome activation by removing the K63-linked ubiquitination of pro-IL-1β. Mechanistically, it was suggested that the removal of ubiquitination on pro-IL-1β reduces its ability to be cleaved by caspase-1 [134,135].

However, a more recent study suggested that K133 ubiquitination of pro-IL-1β can limit inflammasome activation [136]. The study showed that the ubiquitination of pro-IL-1β limits the cellular level of pro-IL-1β by promoting its proteasomal degradation. Moreover, the ubiquitination inhibited IL-1β release through interfering with the caspase-1-mediated pro-IL-1β cleavage [136]. These results indicate that the ubiquitination of pro-IL-1β may play a role in limiting IL-1β level and caspase-1-mediated activation, which is contrary to previous studies. Further investigation is needed to elucidate why ubiquitination at the same residue has opposing effects in different sets of experiments.

## 6. Ubiquitin-Associated NLRP3 Inflammasome Inhibitors

The direct linkage among a wide range of inflammatory diseases and NLRP3 inflammasome complexes makes it a prominent therapeutic target [15]. Despite being known for a long time, the complete and clear regulation of the NLRP3 inflammasome complex is still not within reach. Targeting ubiquitination for the regulation of NLRP3 inflammasome is critical. Although there are several promising NLRP3 inflammasome inhibitors, such as MCC950 [137], Bay-117082 [138], parthenolide [138], glyburide [139], and B-hydroxybutyrate [140], the exact mechanism is still not clear because of the lack of understanding with regard to the regulation of NLRP3 inflammasome activation. Only a small number of NLRP3 inflammasome inhibitors has been shown to target solely the ubiquitination system of NLRP3, as described below (Table 4) [55,102,141,142,143]. 

It is well understood that an abundance of NLRP3 and ASC is required for NLRP3 inflammasome activation [35]. The ubiquitination system regulates the stability and availability of these proteins. Therefore, this system can be targeted to control NLRP3 inflammasome activation.

BC1215

The first compound to show its efficacy against NLRP3 inflammasome activation was BC1215, which was primarily known as an inhibitor of FBXO3, an E3 ligase that increases NLRP3 inflammasome activation. BC1215 inhibits the interaction of FBXO3 with FBXL2, which prevents ubiquitin-mediated FBXL2 degradation. The stabilizing FBXL2 causes the increased ubiquitination of NLRP3 and thereby suppresses its activation [74].

Celastrol

Several species in the Celastraceae family contain the active compound celastrol, a pentacyclic triterpenoid quinone methide that is known for its anti-inflammatory activity. A recent study reported that celastrol acts as an NLRP3 inhibitor by regulating K63-linked NLRP3 ubiquitination. It was demonstrated that celastrol binds directly to BRCC3, a NLRP3 DUB that prevents further deubiquitination of NLRP3 and the formation of an NLRP3 inflammasome complex. An in vivo study demonstrated that celastrol inhibits K63-linked ubiquitination in both LPS- and MSU-treated mice joint and liver tissue [142].

Tranilast

Tranilast (N-[30,40-dimethoxycinnamoyl]-anthranilic acid) is known for its anti-allergic activity. A recent study showed that tranilast inhibits NLRP3 inflammasome activation both in vitro and in vivo [143]. In line with this, a later study identified the regulatory mechanism of tranilast in NLRP3 inflammasome inhibition [144]. Mechanistically, tranilast increases K63-linked NLRP3 ubiquitination, which leads to a restriction of NLRP3 oligomerization and NLRP3 inflammasome assembly. Furthermore, the study showed that tranilast blunts the progression of atherosclerosis in low-density lipid receptor and apolipoprotein-deficient mice, which are known as the best experimental models with which to study atherosclerosis [144].

**Table 4 ijms-22-08780-t004:** NLRP3 ubiquitin-regulating inhibitors of the NLRP3 inflammasome.

Compounds	Disease Models	Mechanism	Outcome	Reference
C-1215	U937, THP-1, primary human alveolar macrophages	Inhibits FBXO3, increases E3 ligase FBXL2, and decreases ubiquitin to mediate NLRP3 degradation	Inhibition of NLRP3 inflammasome activation	[74]
Celastrol	BMDM, THP-1, LPS-induced liver injury, MSU-induced gouty arthritis model	Binding of celastrol to BRCC3 prevents K63-linked NLRP3 deubiquitination	Inhibition of NLRP3 inflammasome, alleviation of LPS induced liver damage and MSU induced arthritis	[142]
Tranilast	J774A.1, BMDM, atherosclerosis in Ldr-/- and ApoE-/- mice model	Increased K63-linked polyubiquitination	Inhibition of NLRP3 inflammasome, protection against atherosclerosis	[144]
Zinc	BV2 cells, spinal cord injury mouse model	Induces autophagy and targets NLRP3 for degradation	Inhibition of NLRP3 inflammasome, neuroprotection	[145]
Caffeic acid phenethyl ester (CAPE	BMDM, THP-1 DSS/AOM-induced colon cancer mouse model	Increased interaction between NLRP3 and Cullin 1 and decreased interaction between NLRP3 and CSN5	Inhibition of NLRP3 inflammasome activation, anticancer	[146]
ML323	Mouse primary peritoneal macrophages, folic acid-induced acute tubular necrosis (ATN)	Inhibits UAF1/USP1 and keeps NLRP3 in check	Inhibition of Nlrp3 inflammasome activation	[101]

Zinc

Many studies have shown that zinc exhibits antioxidant and anti-inflammatory effects [147,148]. A recent study found that zinc contains an NLRP3 inflammasome inhibitory effect. The same study demonstrated that zinc inhibits NLRP3 inflammasome activation in BV2 microglial cells and spinal cord injury mouse models. The mechanistic study revealed that zinc induces autophagy and targets NLRP3 for ubiquitination, which leads to its subsequent degradation [145]. 

CAPE

Caffeic acid phenethyl ester (CAPE) is a bioactive compound of propolis from honeybee hives that has antioxidant and anti-inflammatory effects [149]. A recent study found that CAPE inhibits ATP-induced NLRP3 inflammasome activation in BMDM and THP-1 cells [146]. Moreover, CAPE is involved in the protection of mice in the DSS or azoxymethane (AOM)-induced colorectal cancer. The study also showed that CAPE promotes the ubiquitination of NLRP3 through the inhibition of ROS and increases the level of interaction between NLRP3 and the E3 ligase Cullin 1, whereas it decreases the interaction between NLRP3 and CSN5 (a DUB), which together increases NLRP3 ubiquitination. These results were also found in an AOM/DSS mouse model [146]. 

ML323

A recent study identified ML323 as a novel NLRP3 inflammasome inhibitor [101]. The study reported that UAF1/USP1 is a deubiquitinase complex that eliminates the K48-linked polyubiquitination of NLRP3. As K48-linked ubiquitination is linked with protein degradation, the UAF1/USP1 complex stabilizes the degradation of NLRP3, which increases cellular NLRP3 levels and promotes NLRP3 inflammasome activation. Since ML323 has an inhibitory effect on the UAF1/USP1 complex, its treatment keeps NLRP3 in check at the cellular level and prevents NLRP3 inflammasome activation. Moreover, the presence of ML323 in folic acid-induced acute tubular necrosis (ATN) inhibited NLRP3 inflammasome-mediated inflammation [101].

## 7. Conclusions and Future Perspective

Dysregulated or uncontrolled NLRP3 inflammasome activation plays a significant role in the onset of several inflammatory disorders [15]. The PTM of NLRP3 plays a critical role in maintaining a controlled NLRP3 inflammasome activation. Recent advances in this area of research have demonstrated the essential role of the ubiquitin system in controlling the NLRP3 inflammasome [70] (Figure 2). It has been shown that the ubiquitin system is a potential therapeutic target with which to treat different forms of cancer and neurodegenerative disease [150,151]. 

Several ubiquitin proteasome system (UPS) modulators/inhibitors, such as bortezomib, carfilzomib, ixazomib, and marizomib, are being used successfully in cancer treatment or clinical trials [152,153,154,155]. Even though these UPS modulators or inhibitors are a promising strategy for cancer treatment, some side effects and non-target effects still need to be addressed before their wider application [156,157]. 

Ubiquitin-mediated post-translational modification of NLRP3 inflammasome activation is an emerging area of interest to search for therapeutic targets to control NLRP3 inflammasome activation. Small molecules such as MCC950, OLT1177, MNS, CY-09, and BOT4-one have the most potential as small molecular inhibitors of the NLRP3 inflammasome [137,158,159,160,161], and some are currently undergoing preclinical or clinical trials [162]. Some inhibitors have recently been identified to target the ubiquitin-mediated PTM in NLRP3 inflammasome activation, and these inhibitors are also potential targets to control NLRP3 inflammasome activation [101,144]. Targeting specific E3 ligases or DUBs by these inhibitors might be a new prospect in the future of NLRP3 inflammatory disease therapeutics. 

However, the exact mechanism behind the regulation of NLRP3 inflammasome activation through ubiquitination still requires further study. For instance, there is scant knowledge regarding how the ubiquitination of NLRP3 inflammasome components NEK7, ASC, caspase-1, and IL-1β regulates NLRP3 inflammasome complex formation. Additionally, the exact sites or types of ubiquitin chains that are involved in the specific NLRP3 inflammasome component ubiquitination need to be elucidated, and the specific E3 ligases and DUBs that regulate NLRP3 inflammasome activation need to be clarified to make it a therapeutic target.

## Figures and Tables

**Figure 1 ijms-22-08780-f001:**
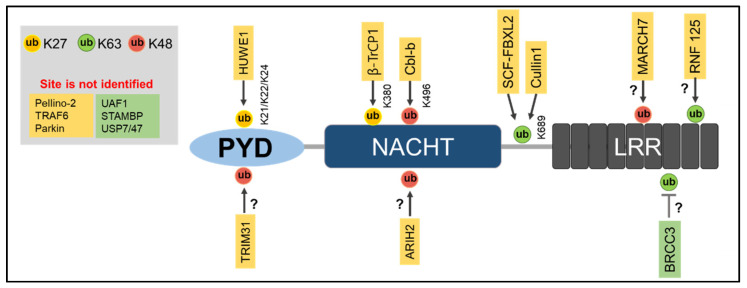
E3 ligases and DUBs involved in ubiquitination and deubiquitination of NLRP3. E3 ligases and DUBs targets different lysine sites of NLRP3 for ubiquitination or deubiquitination. Yellow indicates ubiquitin ligases and green indicates deubiquitinase (DUB).

**Figure 2 ijms-22-08780-f002:**
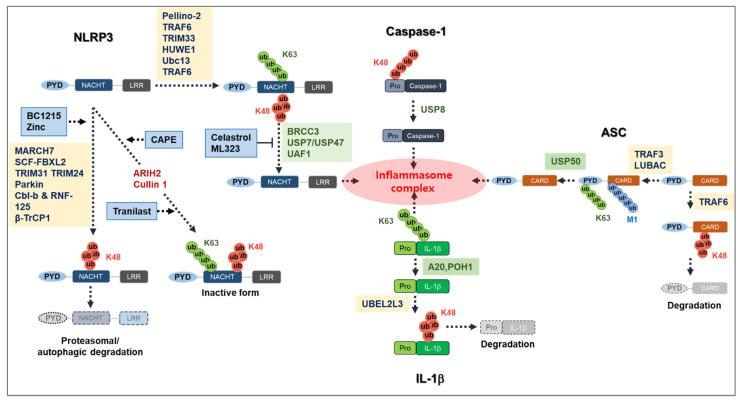
Overview of ubiquitin-mediated regulation of NLRP3 inflammasome activation. Yellow indicates ubiquitin ligases and green indicates deubiquitinase (DUB). Blue indicates the proposed molecules that inhibit NLRP3 inflammasome activation by modulation of ubiquitination.

**Table 1 ijms-22-08780-t001:** Regulation of NLRP3 by E3 ligases.

Name	Regulation of NLRP3 Inflammasome	Impact on NLRP3	Type of UB Linkage	Ubiquitination Site	Interaction	Stage of NLRP3 Activation	Reference
MARCH7	Negative	Autophagy mediated NLRP3 degradation	K48	LRR domain	NACHT, LRR	Active state	[55]
SCF-FBXL2	Negative	NLRP3 degradation	Not identified	K-689(human)	Trp-73 in PYD, Lys689 in LRR	Priming	[71]
TRIM31	Negative	Proteasomal NLRP3 degradation	K48	Not identified	PYD domain	Priming	[68]
ARIH2(Ariadne homolog 2)	Negative	Suppress NLRP3 inflammasome activation	K48/K63	Not identified	NACHT domain	Priming/ Activation	[69]
Cullin 1	Negative	Suppresses NLRP3 inflammasome activation	K63	K-689	PYD, NACHT, and LRR domain	Priming	[72]
Cbl-b	Negative	Proteasomal NLRP3 degradation	K48	K496	LRR domain	Priming/ Activation	[18]
RNF-125	Negative	Recruits Cbl-b and promotes NLRP3 degradation	K63	LRR domain of NLRP3	LRR domain	Priming	[18]
Pellino-2	Positive	Ubiquitinates NLRP3	K63	Not identified	Not identified	Priming	[73]
TRAF6	Positive	Oligomerization of NLRP3 and facilitates NLRP3–ASC interaction	Not identified	Not identified	No interaction with NLRP3	Nontranscriptional priming	[74]
HUWE1	Positive	Promotes inflammasome assembly	K27	K21, K22, and K24	NACHT domain	Activation	[75]
Parkin	Negative	Induces A20, which suppresses NF-κB activation and subsequent NLRP3 activation	Not identified	Not identified	Not identified	Priming	[76]

**Table 3 ijms-22-08780-t003:** Regulation of ASC by E3 ligase and deubiquitinase.

**Name**	**Regulation of Inflammasome Activation**	**Impact on** **ASC**	**Type of** **UB Linkage**	**Ubiquitination Site**	**Stage of NLRP3 Activation**	**Reference**
LUBAC	Positive	Linear ubiquitination	M1	PYD	Priming	[119]
TRAF3	Positive	Ubiquitination at K174	K63	K174	Priming	[122]
TRAF6	Negative	Ubiquitination	K63	Not identified	Not identified	[123]
USP50	Positive	Deubiquitination	Not identified	Not identified	Activation	[124]

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
