# Peer review of "NLRP3 Ubiquitination—A New Approach to Target NLRP3 Inflammasome Activation"

_ijms, 2021, doi:10.3390/ijms22168780_

Round 1

Reviewer 1 Report

This is quite an instructive review of NLRP3 inflammasome ubiquitination and potential uses of the ubiquitin system as a novel means of therapeutically targeting NLRP3 inflammasome activation.

There is an extensive reference section – for completeness suggest including recent review by McKee CM, Coll RC. J Leukoc Biol. 2020;108:937-952 on post-translational modifications of NLRP3 inflammasome.

General comment –

This is a valuable paper with comprehensive list of ubiquitin-mediated regulators of NLRP3 inflammasome activation is provided, with mechanisms and therapeutic uses discussed.  

Specific comments –

 The report is very well written with useful Figures/ Tables - one minor error noted so suggest changing -

Pg 11, line 491 “It is well understood that the abundance of NLRP3 and ASC is required for NLRP3 inflammasome activation [35]” to “It is well understood that an abundance of NLRP3 and ASC is required for NLRP3 inflammasome activation”.

Author Response

This is quite an instructive review of NLRP3 inflammasome ubiquitination and potential uses of the ubiquitin system as a novel means of therapeutically targeting NLRP3 inflammasome activation.

Response: We thank the reviewer for the positive comment.

There is an extensive reference section – for completeness suggest including recent review by McKee CM, Coll RC. J Leukoc Biol. 2020;108:937-952 on post-translational modifications of NLRP3 inflammasome.

Response: We thank the reviewer for providing us an important reference. We have included it (reference#57) at page 3, and 12.

General comment –

This is a valuable paper with comprehensive list of ubiquitin-mediated regulators of NLRP3 inflammasome activation is provided, with mechanisms and therapeutic uses discussed.

Response: We thank the reviewer for the positive comment.

Specific comments –

The report is very well written with useful Figures/ Tables - one minor error noted so suggest changing -

Pg 11, line 491 “It is well understood that the abundance of NLRP3 and ASC is required for NLRP3 inflammasome activation [35]” to “It is well understood that an abundance of NLRP3 and ASC is required for NLRP3 inflammasome activation”.

Response: Sentence has been corrected as the reviewer suggested

Reviewer 2 Report

In this review, Akther and colleagues report how the ubiquitination/deubiquitination of NLRP3 complex modulates the IL1b production and the subsequent inflammatory program. This topic is original because the main part of the NLRP3 regulation occurs in a transcriptional manner upon different stresses. Although this review is in general well-written and of interest for a broad readership, some parts of this review should be improved.

Major points

  1. A brief description of the transcription-dependent NLRP3 activation (signal 1) should be added in this review.
  2. A brief description of the phosphorylation-mediated regulation of NLRP3 should be added to complete the sentence “suggest that a serine-threonine kinase NIMA-related kinase 7 (NEK7) interacts with NLRP3 and form oligomers that are essential for ASC speck formation and caspase 1 activation [47]. “
  3. A figure showing the structure of NLRP3 with the different lysines targeted for ubiquitination/deubiquitination (and the associated E3-ligase and DUB) should be provided.

Minor points

  1. “inflammasome activation. In wild-type bone marrow-derived macrophages (BMDM), pellino-2 ubiquitinates IL-1R associated kinase 4 (IRAK1), which is required for rapid activation of NLRP3 [87]. «

Please, correct in this paragraph: IRAK1 or IRAK4 ?

  1. “Histone deacetylase (HDAC6) exhibits ubiquitin-binding activity and can transport ubiquitinated protein via microtubules [113]. A previous study revealed that HDAC6 can associate with ubiquitinated NLRP3 by its ubiquitin-binding domain and negatively regulates NLRP3 inflammasome activation. Moreover, treatment of a deubiquitinase inhibitor PR619 caused an increased association between HDAC6 and NLRP3 [114]. These results indicate that HDAC6 may negatively regulate NLRP3 inflammasome activation through a direct association with ubiquitinated NLRP3 [114].”

An additional paragraph should be placed in this review because HDAC6 is neither an E2 ligase nor a DUB. For instance, a paragraph designated “Ubiquitinase and DUB-independent regulation of the NLRP3 ubiquitination”

  1. “forms Meteonine1 (Met1)-linked linear ubiquitin chains to the substrate proteins that regulate the classical activation of NF-κB [117].”

Authors should replace “Meteonine1” with “Methionine”.

  1. “The ubiquitination of pro-IL1β by A20 occurs at the lysine 133 site and promotes the proteolytic processing of caspase 1. As a DUB, A20 restricts the ubiquitination of pro-IL1β and contributes to dampening NLRP3 inflammasome activation by inhibiting the A20 dependent proteolytic activity of pro-IL1β [134].”

Authors have to rewrite these two sentences because IL1 is not an enzyme and authors have to better explain how the A20-dependent IL1b ubiquitination promotes the degradation of caspase-1? This sentence is not clear.

Author Response

In this review, Akther and colleagues report how the ubiquitination/deubiquitination of NLRP3 complex modulates the IL1b production and the subsequent inflammatory program. This topic is original because the main part of the NLRP3 regulation occurs in a transcriptional manner upon different stresses. Although this review is in general well-written and of interest for a broad readership, some parts of this review should be improved.

Major points

1) A brief description of the transcription-dependent NLRP3 activation (signal 1) should be added in this review.

Response: We thank the reviewer for providing us with important comments. A subsection “2.1 Priming of NLRP3” has been added at page 2, line 96 describing the transcription-dependent NLRP3 activation (signal 1).

2) A brief description of the phosphorylation-mediated regulation of NLRP3 should be added to complete the sentence“ suggest that a serine-threonine kinase NIMA-related kinase 7 (NEK7) interacts with NLRP3 and form oligomers that are essential for ASC speck formation and caspase 1 activation [47]. “

Response: Following sentence has been added at page 3, line 122 to explain NEK 7 phosphorylation mediated regulation of NLRP3.

“A recent study showed that this NEK7-NLRP3 interaction can be regulated by the phos-phorylation of NEK7. The phosphorylation of NEK7 occurs at Ser204 by polo like kinase 4 (PLK4) upon LPS stimulation [50]. Since the phosphorylation attenuates the association of NEK7 with NLRP3, it might serve to restrain inflammasome activation [50].”

3) A figure showing the structure of NLRP3 with the different lysines targeted for ubiquitination/deubiquitination (and the associated E3-ligase and DUB) should be provided.

Response: We thank the reviewer for important comment.

A figure has been added on page 5, as per recommendation explaining different lysines targeted for ubiquitination/deubiquitination (and the associated E3-ligase and DUB).

 Minor points

1) “inflammasome activation. In wild-type bone marrow-derived macrophages (BMDM), pellino-2 ubiquitinates IL-1R associated kinase 4 (IRAK1), which is required for rapid activation of NLRP3 [87]. Please, correct in this paragraph: IRAK1 or IRAK4 ?    

Response: Thank you for pointing out. It was an erratum and has been corrected.

 2) “Histone deacetylase (HDAC6) exhibits ubiquitin-binding activity and can transport ubiquitinated protein via microtubules [113]. A previous study revealed that HDAC6 can associate with ubiquitinated NLRP3 by its ubiquitin-binding domain and negatively regulates NLRP3 inflammasome activation. Moreover, treatment of a deubiquitinase inhibitor PR619 caused an increased association between HDAC6 and NLRP3 [114]. These results indicate that HDAC6 may negatively regulate NLRP3 inflammasome activation through a direct association with ubiquitinated NLRP3 [114].”  An additional paragraph should be placed in this review because HDAC6 is neither an E2 ligase nor a DUB. For instance, a paragraph designated “Ubiquitinase and DUB-independent regulation of the NLRP3 ubiquitination”

Response: We thank the reviewer for the suggestion.

A paragraph has been created at page 9, line 414 as 5.1.3, as the reviewer suggested “Ubiquitinase and DUB-independent regulation of the NLRP3 ubiquitination” and under this paragraph regulation of NLRP3 by HDAC6 has been included.

 3) “forms Meteonine1 (Met1)-linked linear ubiquitin chains to the substrate proteins that regulate the classical activation of NF-κB [117].” Authors should replace “Meteonine1” with “Methionine”.

Resopnse: The typo has been corrected now.

4) “The ubiquitination of pro-IL1β by A20 occurs at the lysine 133 site and promotes the proteolytic processing of caspase 1. As a DUB, A20 restricts the ubiquitination of pro-IL1β and contributes to dampening NLRP3 inflammasome activation by inhibiting the A20 dependent proteolytic activity of pro-IL1β [134].”

Authors have to rewrite these two sentences because IL1 is not an enzyme and authors have to better explain how the A20-dependent IL1b ubiquitination promotes the degradation of caspase-1? This sentence is not clear.

Response: We thank the reviewer for the helpful comments. Caspase-1 was a typo and unintentional error in the manuscript. We have rewritten the sentences as the reviewer suggested and included recent publication related to pro-IL-1b ubiquitination.

“Other study focused on the impact of A20, a deubiquitin-modifying enzyme, on the in-flammasome activation[136]. This study demonstrated that the depletion of A20 in mac-rophages results in spontaneous IL-1b processing and release in response to priming acti-vation. Further mechanistic studies showed that ubiquitination at K133 of mouse pro-IL-1β supports the cleavage of pro-IL-1β and A20 limits this proteolytic cleavage through the re-striction of pro-IL-1b ubiquitination [136].

In line with this notion, another deubiquitinase, POH1, is also reported to restrict NLRP3 inflammasome activation by removing the K63-linked ubiquitination of pro-IL-1β. Mechanistically, it was suggested that the removal of ubiquitination on pro-IL-1b reduced its ability to be cleaved by caspase-1[137, 138].  

However, a more recent study suggested that K133 ubiquitination of pro-IL-1β can limits inflammasome activation [139]. The study showed that the ubiquitination of pro-IL-1β limits the cellular level of pro-IL-1β by promoting its proteasomal degradation. Moreover, the ubiquitination inhibited IL-1β release through interfering the caspase-1-mediated pro-IL-1β cleavage [139]. These results indicate that the ubiquitination of pro-IL-1b may play a role in limiting IL-1b level and caspase-1-mediated activation, which is contrary to previous studies. Further investigation is needed to elucidate why ubiquitination at the same residue has opposing effects in different sets of experiments.”

Round 2

Reviewer 2 Report

The authors have addressed all my concerns. Nice piece of work.